# CausalXtract, a flexible pipeline to extract causal effects from live-cell time-lapse imaging data

Franck Simon[1†], Maria Colomba Comes[2†], Tiziana Tocci[1,2†], Louise Dupuis[1], Vincent Cabeli[1], Nikita Lagrange[1], Arianna Mencattini[2], Maria Carla Parrini[3], Eugenio Martinelli[2*], Herve Isambert[1*]

[1]CNRS UMR168, Institut Curie, Université PSL, Sorbonne Université, Paris, France; [2]Department of Electronic Engineering, University of Rome Tor Vergata, Rome, Italy; [3]INSERM U830, Institut Curie, Université PSL, Paris, France

*For correspondence:
martinelli@ing.uniroma2.it (EM);
herve.isambert@curie.fr (HI)

[†]These authors contributed equally to this work

Competing interest: The authors declare that no competing interests exist.

## eLife Assessment

This **important** study represents a data processing pipeline to discover causal interactions from time-lapse imaging data and **convincingly** illustrates it on a challenging application for the analysis of tumor-on-chip ecosystem data. The authors describe the raw data they used (imaging data), go through a step-by-step description of how to extract the features they are interested in from the raw data, and how to perform the causal discovery process. This article tackles the problem of learning causal interactions from temporal data, which is applicable to many biological applications.

**Abstract** Live-cell microscopy routinely provides massive amounts of time-lapse images of complex cellular systems under various physiological or therapeutic conditions. However, this wealth of data remains difficult to interpret in terms of causal effects. Here, we describe CausalXtract, a flexible computational pipeline that discovers causal and possibly time-lagged effects from morphodynamic features and cell–cell interactions in live-cell imaging data. CausalXtract methodology combines network-based and information-based frameworks, which is shown to discover causal effects overlooked by classical Granger and Schreiber causality approaches. We showcase the use of CausalXtract to uncover novel causal effects in a tumor-on-chip cellular ecosystem under therapeutically relevant conditions. In particular, we find that cancer-associated fibroblasts directly inhibit cancer cell apoptosis, independently from anticancer treatment. CausalXtract uncovers also multiple antagonistic effects at different time delays. Hence, CausalXtract provides a unique computational tool to interpret live-cell imaging data for a range of fundamental and translational research applications.

## Introduction

Live-cell imaging microscopy commonly produces extensive amounts of time-lapse images of cellular systems, which can be segmented to extract morphodynamic features and interactions of individual cells under increasingly complex and physiologically relevant conditions. However, this wealth of information remains largely underexploited due to a lack of methods and tools able to discover causal effects from spatio-temporal correlations under well-controlled experimental conditions.

CausalXtract addresses this need by integrating an advanced live-cell image feature extraction tool with a reliable and scalable causal discovery method (***Figures 1 and 2***) in order to learn temporal causal networks from live-cell time-lapse imaging data (***Figure 3***).

**Figure 1.** CausalXtract pipeline. (**a**) Live-cell tumor ecosystem reconstituted ex vivo (*Nguyen et al., 2018*) using the tumor-on-chip technology ('Materials and methods'). (**b**) CausalXtract's live-cell image feature extraction module (CellHunter+). The tracking of cancer and immune cells and of their mutual interactions is illustrated in *Videos 1–3*, in the absence or presence of cell division and apoptosis event. Examples of time series of extracted cellular features are shown in *Figure 1—figure supplement 1*. (**c**) CausalXtract's temporal causal discovery module (tMIIC) learns a temporal causal network from the features extracted in (**b**). See 'Materials and methods' for CausalXtract's implementation details and theoretical foundations. A step-by-step notebook of CausalXtract pipeline is provided with the source code.

The online version of this article includes the following figure supplement(s) for figure 1:

**Figure supplement 1.** Time series of cellular features extracted from the tumor ecosystems.

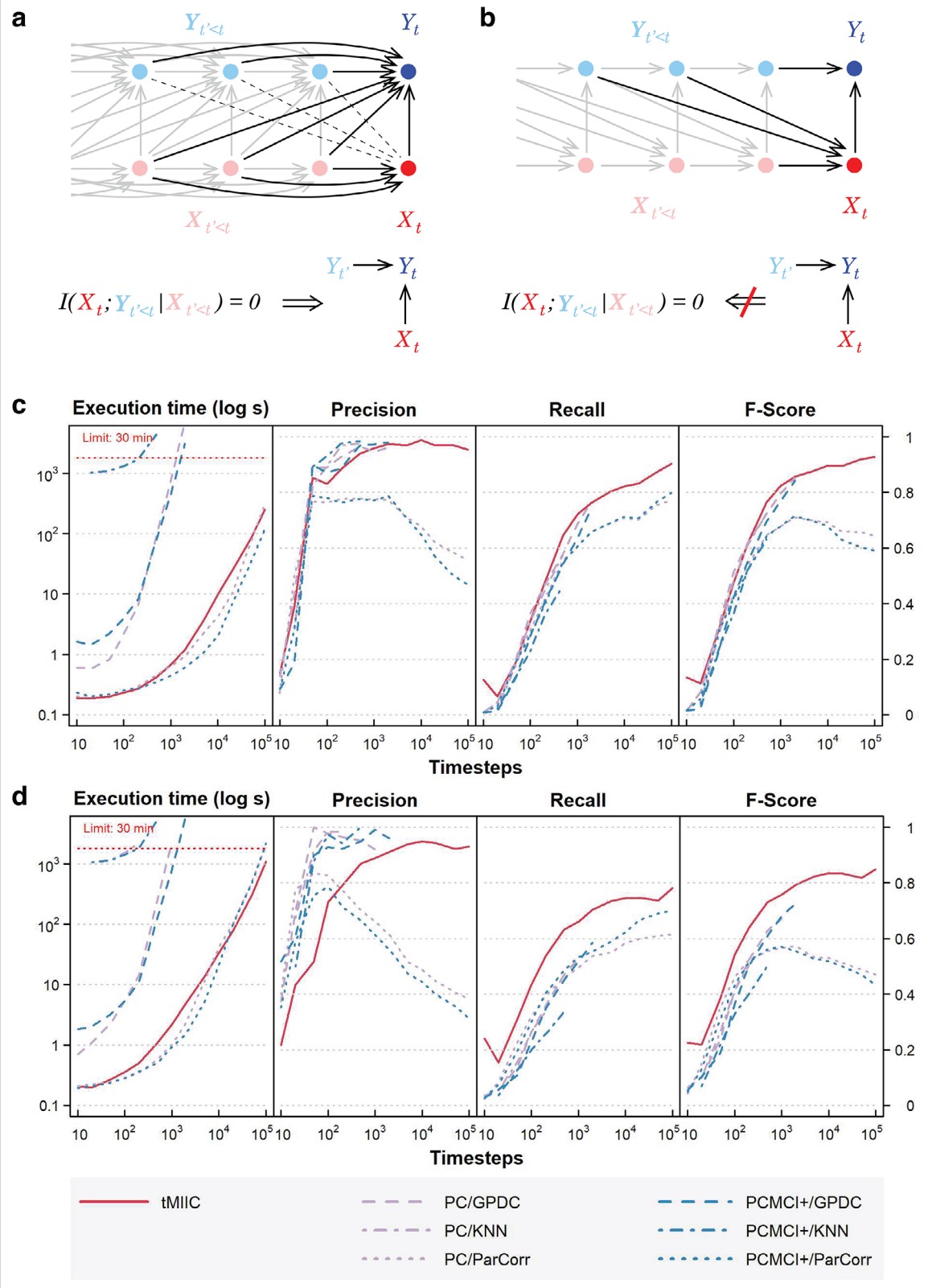

**Figure 2.** Relation to Granger–Schreiber temporal causality and tMIIC benchmarking against PC and PCMCI+. (**a**) The signature of Granger–Schreiber temporal causality is a vanishing Transfer Entropy, that is, $T_{Y \to X} = I(X_t; Y_{t' < t} | X_{t' < t}) = 0$ ('Materials and methods'). In the time-unfolded causal network framework, it implies (i) the absence of (dashed) edge between $X_t$ and any $Y_{t'}$, with $t' < t$, and (ii) if $X_t$ is adjacent to $Y_t$, the presence of temporal (2-variable+time) v-structures, $Y_{t'} \to Y_t \leftarrow X_t$, for all $Y_{t'}$ adjacent to $Y_t$, with $t' < t$ ('Materials and methods', Theorem 1). (**b**) By contrast, the presence

*Figure 2 continued on next page*

*Figure 2 continued*

of a temporal (2-variable+time) v-structure, $Y_{t'} \rightarrow Y_t \leftarrow X_t$ does not imply a vanishing Transfer Entropy as long as there remains an edge between any $Y_{t'' < t}$ and $X_t$. It implies that Granger–Schreiber temporal causality is in fact too restrictive and may overlook actual causal effects, which can be uncovered by graph-based causal discovery methods. Hence, tMIIC's time-unfolded network framework, combining graph-based and information-based approaches, sheds light on the common foundations of the seemingly unrelated graph-based causality and Granger–Schreiber temporal causality, while clarifying their actual differences and limitations. (**c**) Benchmarking of tMIIC on synthetic time-series datasets generated from 15-node causal networks based on linear combinations of contributions, Appendix 1 and *Figure 2—figure supplements 1–3*. (**d**) Benchmarking with more complex 15-node time-series datasets based on nonlinear combinations of contributions, Appendix 2 and *Figure 2—figure supplement 4*. Running times and scores (Precision, Recall, Fscore) are averaged over 10 datasets and compared to PC and PCMCI+ methods using different kernels (GPDC, KNN, ParCorr).

The online version of this article includes the following figure supplement(s) for figure 2:

**Figure supplement 1.** Benchmark assessment of CausalXtract's causal discovery module (tMIIC) using generated time-series datasets.

**Figure supplement 2.** CausalXtract insensitivity to an overestimated maximum lag $\tau$.

**Figure supplement 3.** CausalXtract sensitivity to non-stationary variables.

**Figure supplement 4.** Benchmark assessment of CausalXtract's causal discovery module (tMIIC) using more complex time-series datasets.

## Results

### CausalXtract's feature extraction and causal discovery modules

CausalXtract's live-cell image feature extraction module (CellHunter+) (*Figure 1b*) is based on Cell-Hunter software (*Nguyen et al., 2018*) and consists of three steps: detection, tracking, and feature extraction of live cells within time-lapse video images. First, automatic localization/segmentation of cells (e.g., tumor and immune cells) is performed with the Circular Hough Transform (CHT) algorithm (*Davies, 2004*) to estimate the cell centers and radii. Second, cell trajectories along the frames are constructed by linking the positions detected at the previous time step through Munkres' algorithm for optimal sub-pattern assignment problems (OAPs) (*Munkres, 1957*). Finally, relevant descriptors related to the shape, motility, and state of the cells, as well as cell–cell interactions, are quantified from each cell trajectory ('Materials and methods').

CausalXtract's temporal causal discovery module (tMIIC) (*Figure 1c*) is adapted from the causal discovery method (MIIC) (*Verny et al., 2017*; *Cabeli et al., 2020*; *Cabeli and Li, 2021*; *Ribeiro-Dantas et al., 2024*), which learns contemporaneous causal networks (i.e., when temporal information is not available) for a broad range of biological or biomedical data, from single-cell transcriptomic and genomic alteration data (*Verny et al., 2017*; *Desterke et al., 2020*) to medical records of patients (*Cabeli et al., 2020*; *Sella et al., 2022*; *Ribeiro-Dantas et al., 2024*). Live-cell time-lapse imaging data contain, however, information about cellular dynamics, which can in principle facilitate the discovery of novel cause–effect functional processes based on the assumption that future events cannot cause past ones. To this end, CausalXtract's discovery module, tMIIC, reconstructs time-unfolded causal networks, where each variable is represented by several nodes at different relative time points (*Assaad et al., 2022*; *Figure 1c*). Such a time-unfolded network framework (*Entner and Granger, 2010*; *Malinsky and Spirtes, 2018*; *Runge et al., 2019*) is required to account for the temporal correlation between successive time steps in time-series data. This graph-based causal framework goes beyond the seminal concept of temporal causality originally proposed by *Granger, 1969* for linear time series without reference to graphical models and later extended to nonlinear dynamics by *Schreiber, 2000*; *Barnett et al., 2009*. In particular, Granger–Schreiber causality is in fact too restrictive and may overlook actual causal effects that can be uncovered by graph-based causal discovery methods (*Figure 2a and b*; 'Materials and methods', Theorem 1). In addition, Granger–Schreiber causality has long been known to infer spurious causal associations based on time delays by excluding the presence of latent common causes a priori (*Assaad et al., 2022*). tMIIC circumvents these limitations by combining graph-based and information-based approaches ('Materials and methods'), while including contemporary and time-delayed effects of unobserved latent variables that are ubiquitous in cell biology data (e.g., the latent effects of cell cycle phases on cellular features and responses).

We benchmarked tMIIC on synthetic datasets resembling the real-world data of interest analyzed in this study (i.e., number of time steps, network size, and degree distribution) and found that it matches or outperforms state-of-the-art methods, PC and PCMCI+ (*Runge, 2020*), while running order of magnitudes faster on datasets of biologically relevant size including tens to hundreds of thousands

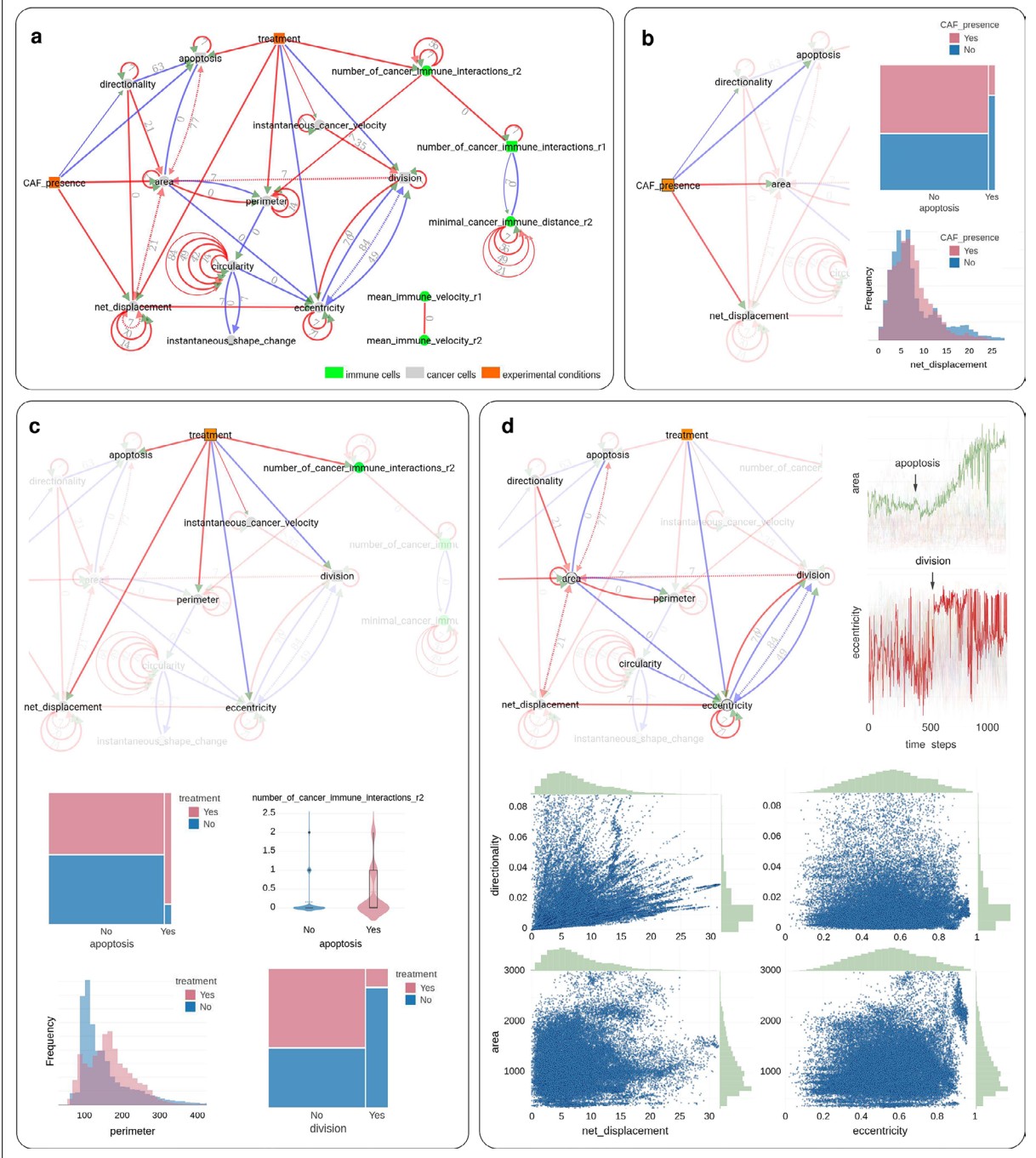

**Figure 3.** Application of CausalXtract to time-lapse images of tumor ecosystems reconstituted ex vivo. (**a**) Summary causal network inferred by CausalXtract. The underlying time-unfolded causal network is shown in *Figure 3—figure supplement 1*. Red (resp. blue) edges correspond to positive (resp. negative) associations. Bidirected dashed edges represent the effect of unobserved (latent) common causes. Annotations on edges correspond to time delays in time steps (1 ts = 2 min). The inferred network is largely robust to variations in sampling rate ($\delta\tau$) and maximum lag ($\tau$), *Figure 3—figure supplement 2*. Here, $\delta\tau = 7$ ts and $\tau = 84$ ts are chosen automatically by CausalXtract. (**b**) The CAF presence subnetwork highlighting the direct causal effects of CAFs on cancer cells. In particular, CausalXtract uncovers that CAFs directly inhibit cancer cell apoptosis independently from treatment, which has not been reported so far. (**c**) The treatment subnetwork highlighting the direct causal effects of treatment on cancer cells. In particular, CausalXtract uncovers that treatment increases cancer cell perimeter, which has not been reported either. (**d**) The eccentricity-area subnetwork highlighting multiple direct and possibly antagonistic time-lagged effects, notably, between cell division and eccentricity and between cell apoptosis and area, as discussed in the main text.

The online version of this article includes the following figure supplement(s) for figure 3:

*Figure 3 continued on next page*

**Figure supplement 1.** Time-unfolded causal network inferred by CausalXtract.

**Figure supplement 2.** Robustness of CausalXtract's temporal causal networks to variations in sampling rate.

time steps (*Figure 2c and d* and *Figure 2—figure supplement 3; Figure 2—figure supplement 2; Figure 2—figure supplement 1; Figure 2—figure supplements 4*).

## Application to tumor-on-chip cellular ecosystems

We showcase CausalXtract with the analysis of time-lapse images of a tumor ecosystem reconstituted ex vivo using the tumor-on-chip technology (*Figure 1a*). These live-cell time-lapse images come from a proof-of-concept study (*Nguyen et al., 2018*), which demonstrated the effects of an anticancer drug (the monoclonal antibodies trastuzumab, brand name Herceptin, used to treat HER2+ breast cancers) on a reconstituted tumor microenvironment, including cancer cells, immune cells, cancer-associated fibroblasts (CAFs), and endothelial cells ('Materials and methods'). However, a comprehensive extraction and analysis of cellular morphodynamic features and interactions remained unexplored.

To this end, cellular features such as cell geometry, velocity, division, apoptosis, cell–cell transient interactions, and persistent contacts were first extracted from the raw images using CausalXtract's feature extraction module (*Figure 1b*, *Figure 1—figure supplement 1*).

Then, summary causal network (*Figure 3a*) and the corresponding time-unfolded causal network (*Figure 3—figure supplement 1*) were reconstructed between extracted cellular features, cell–cell interactions, and therapeutic conditions using CausalXtract's temporal causal discovery module (*Figure 1c*).

CausalXtract inferred network (*Figure 3a*) uncovers novel biologically relevant findings, in addition to confirming known results from earlier studies. In particular, CausalXtract discovers that CAFs directly inhibit cancer cell apoptosis, independently from anticancer treatment (*Figure 3b*), while earlier studies reported that CAFs merely reduced the effect of treatment (*Nguyen et al., 2018*). CausalXtract also discovers that treatment increases cancer cell perimeter (*Figure 3c*), which has not been reported so far either. In addition, CausalXtract confirms known results from earlier studies. In particular, it recovers that treatment increases cancer cell apoptosis and the number of cancer-immune interactions, as well as decreases the division rate of cancer cells (*Figure 3c*). Likewise, CausalXtract recovers that CAFs stimulate cancer cell migration and increase their area (*Figure 3b*).

Interestingly, CausalXtract identifies also multiple and possibly antagonistic effects with different time delays. For instance, CausalXtract recovers several antagonistic relations between morphodynamic features such as cell division and eccentricity

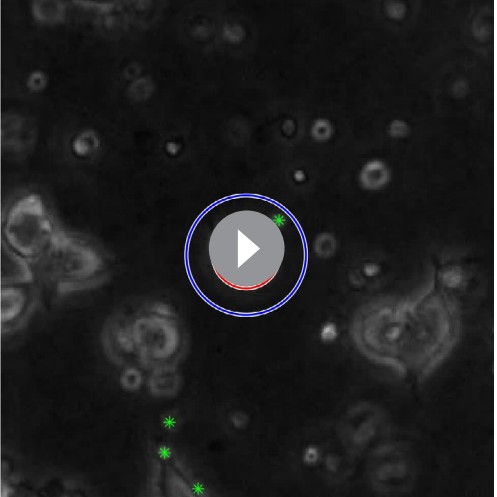

**Video 1.** Example of tracking of cancer and immune cells and of their mutual interactions in the absence of cell division and apoptosis event.

https://elifesciences.org/articles/95485/figures#video1

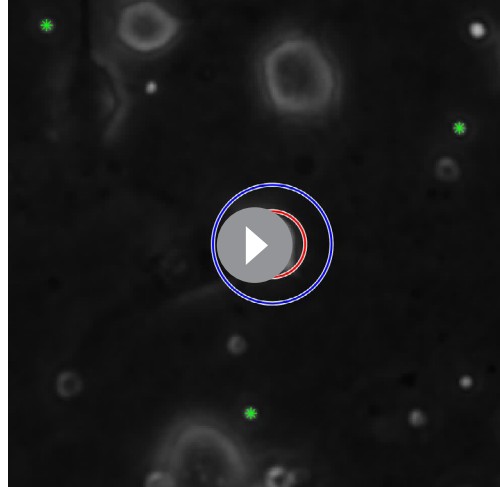

**Video 2.** Example of tracking of cancer and immune cells and of their mutual interactions in the presence of a cell division event.

https://elifesciences.org/articles/95485/figures#video2

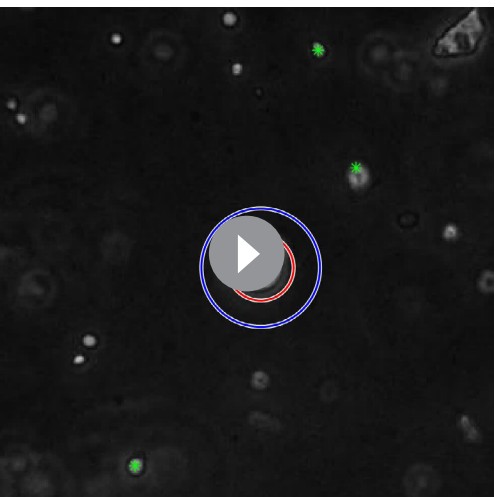

**Video 3.** Example of tracking of cancer and immune cells and of their mutual interactions in the presence of a cell apoptosis event.

https://elifesciences.org/articles/95485/figures#video3

or cell apoptosis and area (*Figure 3d*). Indeed, the late phases of cell division are associated to a marked increase in eccentricity (red edge) but preceded by a net decrease in eccentricity, 2–3 hr before cytokinesis (blue edges), once the decision to divide has been made (i.e., the probable latent cause) and the cell is actually duplicating its biological materials (prophase) (*Figure 3d*). Likewise, the area change upon apoptosis is predicted to first decrease soon after apoptosis (blue edge) before eventually increasing upon cell lysis (red edge) (*Figure 3d*). These results are robust to variations in sampling rate (*Figure 3— figure supplement 2*).

## Discussion

All in all, CausalXtract is a flexible pipeline that uncovers novel and possibly time-lagged causal relations between cellular features under controlled conditions (e.g., drug). CausalXtract uniquely combines live-cell feature extraction with information theory and causal discovery approaches. It consists of two independent computational modules, conceived to warrant interoperability with alternative live-cell segmentation and tracking methods or alternative temporal causal discovery methods.

CausalXtract opens up new avenues to analyze live-cell imaging data for a range of fundamental and translational research applications, such as the use of tumor-on-chips to screen immunotherapy responses on patient-derived tumor samples. With the advent of virtually unlimited live-cell image data, flexible hypothesis-free interpretation methods are much needed (*Driscoll and Zaritsky, 2021*), and we believe that CausalXtract can bring unique insights based on causal discovery to interpret such information-rich live-cell imaging data.

## Materials and methods
### Tumor-on-chip preparation and live-cell microscopy
The videos analyzed in the present study refer to biological experiments emulating a 3D breast tumor ecosystem (*Nguyen et al., 2018*). All tumor-on-chip experiments have a central endothelium compartment containing endothelial cells (primary human umbilical vein endothelial cells [HUVECs]) and two lateral chambers filled with biomimetic hydrogel (collagen type I at 2.3 mg/mL) seeded with cancer cells (HER2+ breast cancer BT474 cell line) and immune cells (peripheral blood mononuclear cells [PBMCs]) from healthy donors (*Figure 1a*). Four experimental conditions were considered depending on the presence or absence of breast CAFs (CAF cell line Hs578T) and drug treatment (trastuzumab, Herceptin). The immortalized human BT474 and Hs578T cell lines were purchased from ATCC (#HTB-20, #HTB-126) and authenticated by SRT profiling (GenePrint 10 system, Promega, #B9510). The human primary HUVEC lines were purchased from Lonza (#C2517A). PBMCs were routinely isolated from the fresh blood of healthy donors by density gradient centrifugation. All cells were periodically tested to exclude mycoplasma contamination using a qPCR-based method (VenorGem Classic, BioValley, #11-1250). Videos were acquired using inverted motorized Leica microscopes with a frame rate of 2 min for up to 48 hr (1440 frames). *Figure 1b* shows a crop frame with cancer cells, PBMCs, and CAFs. Each video was cropped into multiple small 300 × 300 pixel videos (referred to as crops in the following), each of which represented a field of view at subsequent time frames containing a 'main' cancer cell (MCC) initially placed at the center of the image, some PBMC immune cells, other cancer cells, and possibly CAFs within the surrounding of the MCC depending on the experimental

conditions. Thirty-six video crops of up to 1440 frames were analyzed (46,935 frames in total) corresponding to nine video crops per experimental conditions.

## CausalXtract's live-cell image feature extraction module

The live-cell image feature extraction module (CellHunter+) (*Figure 1b*) extends the CellHunter software (*Nguyen et al., 2018*) and consists of three steps: detection, tracking, and feature extraction of live cells within time-lapse video images. First, cell detection is based on the segmentation of circular-shaped objects using CHT (*Davies, 2004*) with radii set around the theoretical radii of the two cell populations ($r_{im} = 4$ px for immune cells and $r_{ca} = 14$ px for MCCs with a pixel resolution $1\,\text{px} = 0.645\,\mu\text{m}$; *Nguyen et al., 2018*). Then, cell tracking is performed by linking cells detected at the $i^{th}$ frame to cells located at the $(i+1)^{th}$ frame within a maximum distance from the detected cell candidate. While the motions of both MCCs and immune cells resemble random walks with time-varying drift and volatility, these two cell types exhibit different motility characteristics (*Nguyen et al., 2018*). Hence, different maximum distances are considered for the two cell populations: it was set to 40 px for MCCs and to 20 px for immune cells. For each cell population, an OAP using the Munkres algorithm *Munkres, 1957* is solved: the globally best possible pairing among located objects is based on an assignment cost equal to the inverse of the distance between pairs of cell candidates at the $i^{th}$ and $(i+1)^{th}$ frames. Cell appearing/disappearing and cell overlaps due to projection errors of the 3D scene in the 2D domain are also handled. Finally, cellular morphodynamic features and cell–cell interaction features are extracted at successive positions along each trajectory. For each MCC, 15 descriptors were extracted (*Figure 1—figure supplement 1*) and classified into four main categories: cell shape, motility, state, and interaction descriptors.

### Shape descriptors

The active contour algorithm implemented in MATLAB (*Chan and Vese, 2001*) was used to segment the MCC boundaries on each video crop frame. Taking as input a frame representing the $i^{th}$ snapshot of the $t^{th}$ MCC, it returns a binary image, where the MCC is represented by a white region. From the binary image, the shape properties of the region occupied by each MCC were extracted using the MATLAB *regionprops* algorithm. The resulting descriptors of the extracted shape are listed below:

- Area indicates the number of pixels composing the region. The equivalent diameter of the $t^{th}$ MCC in the $i^{th}$ frame is defined as $d_i^t = \sqrt{4 \cdot area/\pi}$.
- Perimeter represents the distance along the MCC boundary.
- Circularity is defined as $4 \cdot area \cdot \pi/perimeter^2$, which is equal to 1 when the region is perfectly circular.
- Eccentricity denotes the eccentricity of the ellipse with the same second moments as the region. The value is equal to 1 when the region is a line and to 0 when the region is a circle.
- Instantaneous shape change is defined as $|d_i^t - d_{i-1}^t|$, corresponding to the difference in absolute value of the equivalent diameters between the $i^{th}$ and $(i-1)^{th}$ frames of the $t^{th}$ MCC.

### Motility descriptors

The positions $p_i^t = (x_i^t, y_i^t)$ and $p_{i-1}^t$ of the $t^{th}$ MCC in the $i^{th}$ and $(i-1)^{th}$ frames were compared using the Euclidean distance $d(\cdot)$ to define the following motility parameters:

- Instantaneous cancer velocity (*Masuzzo et al., 2016*) is defined as $d(p_i^t, p_{i-1}^t)/\Delta t$, where $\Delta t$ is the time interval between two consecutive frames.
- Net displacement (*Masuzzo et al., 2016*) indicates the resultant distance between the initial and current positions of the $t^{th}$ MCC, $d(p_1^t, p_i^t)$.
- Directionality (*Masuzzo et al., 2016*) is defined as the ratio of net displacement, $d(p_1^t, p_i^t)$, and curvilinear distance, $\sum_{k=2}^{i} d(p_k^t, p_{k-1}^t)$. It measures the persistence of motion and ranges from 0 for confined cells to 1 for cells moving perfectly straight in one direction.

### State descriptors

They record apoptosis or division events:

- Apoptosis indicates if the MCC has died during the experiment. It is set to 'No' as long as the cell has not died and becomes 'Yes' for the remaining frames after the cell undergoes apoptosis.
- Division indicates if the MCC has divided during the experiment. It is set to 'No' as long as the cell has not divided and becomes 'Yes' for the remaining frames after the cell divides.

### Interaction descriptors

Interactions between MCCs and immune cells were defined with respect to two radii around each MCC, $r_1 = r_{im} + r_{ca} + 2 = 20$ px and $r_2 = 2 \times (r_{im} + r_{ca}) = 36$ px (*Nguyen et al., 2018*). Hence, $r_1$ refers to MCC and immune cells in actual physical contact, while $r_2$ refers to MCC and immune cells in close vicinity. Then, for each sample the following interaction features were defined:

- Number of cancer-immune interactions ($r_2$) corresponds to the number of immune cells within the interaction radius $r_2$ around the MCC on that frame.
- Number of cancer-immune interactions ($r_1$) corresponds to the number of immune cells in close contact with the MCC on that frame.
- Minimal cancer-immune distance ($r_2$) is the minimum distance between the MCC and the immune cells within a radius $r_2$.
- Mean immune velocity ($r_2$) is the mean instantaneous velocity norm of the immune cells within the interaction radius $r_2$ around the MCC.
- Mean immune velocity ($r_1$) is the mean instantaneous velocity norm of the immune cells in close contact with the MCC.

## Overview of causal discovery methods for non-temporal data

Traditional causal discovery methods (*Pearl, 2009*; *Spirtes, 2000*) aim to learn causal networks from datasets of independent samples by proceeding through successive steps. They first learn structural constraints in the form of unconditional or conditional independence between variables and remove the corresponding edges from an initial fully connected network. The second step then consists of orienting some of the retained edges based on the signature of causality in observational data. This corresponds to orienting three-variable 'v-structure' motifs as $X \rightarrow Z \leftarrow Y$ whenever the edge $X - Y$ has been removed without conditioning on the variable $Z$, which implies that $Z$ cannot be a cause of $X$ nor $Y$. This does not guarantee, however, that $X$ (or $Y$) is an actual cause of $Z$, which also requires to rule out the possibility that the edge between $X$ and $Z$ (or $Y$ and $Z$) might originate from a latent common cause, $L$, unobserved in the dataset, that is, $X \leftarrow\!- L -\!\rightarrow Z$. In addition, classical causal discovery methods are prone to spurious conditional independences, which lead to many false-negative edges and limit the accuracy of inferred orientations. The recent causal discovery method (MIIC) (*Verny et al., 2017*; *Cabeli et al., 2020*; *Cabeli and Li, 2021*; *Ribeiro-Dantas et al., 2024*), which combines constraint-based and information-based principles, learns more robust causal graphical models by first collecting iteratively significant information contributors before assessing conditional independences (*Affeldt and Isambert, 2015*; *Affeldt et al., 2016*). In practice, MIIC's strategy limits spurious conditional independences, which improves its edge sensitivity and orientation reliability compared to traditional constraint-based methods. In addition, MIIC can handle missing data (*Cabeli et al., 2020*) and also heterogeneous multimodal data by analyzing continuous and categorical variables on the same footing based on a mutual information supremum principle for finite dataset (*Cabeli et al., 2020*; *Cabeli and Li, 2021*; *Ribeiro-Dantas et al., 2024*). Last, MIIC distinguishes genuine causal relations from putative and latent causal effects (*Ribeiro-Dantas et al., 2024*) that are ubiquitous in real-world applications.

## CausalXtract's causal discovery module for time-series data (tMIIC)

In order to analyze time-series datasets, CausalXtract's causal discovery module (tMIIC) aims to learn a time-unfolded graph, $\mathcal{G}_t$, where each variable is represented by a series of nodes associated with its value at different relative time points (*Figure 1c*). Such a time-unfolded network framework (*Entner and Granger, 2010*; *Malinsky and Spirtes, 2018*; *Runge et al., 2019*) is required to account for the temporal correlation between successive samples in time-series data. Assuming that the dynamics can be considered stationary (see section 'Benchmarking of CausalXtract's causal discovery module'), the time-unfolded graph, $\mathcal{G}_t$, should be translationally invariant over time and can be assigned a periodic structure a priori. In addition, $\mathcal{G}_t$ can be restricted to a few time steps from the running time, $t$, back

to a maximum time lag, $t - \tau$, since nodes at future time points ($t' > t$) cannot a priori influence the observed data at current or previous time points ($t' \leqslant t$) (**Figure 1c**). The maximum time lag $\tau$ should be chosen so as to have little effect on the final graphical model, which can be achieved for instance by setting $\tau$ to twice the average relaxation time of the variables of the dataset. In practice, we may also limit the number of time points $\nu$ in $\mathcal{G}_t$ by introducing a time increment $\delta\tau$ between consecutive time points, which leads to $\nu = \tau/\delta\tau$ time-lagged layers in $\mathcal{G}_t$.

Such a compact periodic graphical representation over a sliding temporal window is learned with tMIIC, which extends MIIC causal discovery method to analyze time-series data. First, tMIIC identifies all necessary edges involving at least one contemporaneous node at time $t$ (**Figure 1c**). Once these time-lagged and contemporaneous necessary edges have been identified, they are simply duplicated at earlier time points to enforce the translational invariance of $\mathcal{G}_t$ skeleton. Time-lagged edges are then pre-oriented with a first arrowhead pointing toward the future, considering that current time points cannot cause earlier events. Then, contemporaneous and time-lagged edges can be further oriented using MIIC orientation probability scores applied to $\mathcal{G}_t$, which may also uncover a second arrowhead (backward in time) for time-lagged edges. This corresponds to time-lagged latent causal effects from unobserved common causes (**Figure 1c**).

Learning such structural models including latent variables from time-series data was first proposed for time-lagged effects (**Entner and Granger, 2010**) and subsequently extended to contemporaneous effects (**Malinsky and Spirtes, 2018**) by adapting the constraint-based FCI method allowing for latent variables (**Spirtes, 2000**). While traditional constraint-based methods suffer from poor recall, the recent PCMCI (**Runge et al., 2019**)/PCMCI+ (**Runge, 2020**) method improves recall by introducing ad hoc conditioning rules for autocorrelated time series. By contrast, tMIIC does not require any ad hoc conditioning rules as it relies on the same robust information-theoretic strategy as MIIC to limit spurious independence and improve edge recall. tMIIC also captures time-lagged and contemporaneous effects due to latent variables.

## Relation to Granger–Schreiber temporal causality

The concept of temporal causality was originally formulated by **Granger, 1969** without reference to any graphical model by comparing linear autoregression with or without past values of possible causal variables. This was later extended to nonlinear relations by **Schreiber, 2000**; **Barnett et al., 2009** using the notion of Transfer Entropy, $T_{X \to Y}$, which can be expressed in terms of multivariate conditional information:

$$T_{X \to Y} = I(Y_t; X_{t'<t} | Y_{t'<t}) \tag{1}$$

where $X_{t'<t}$ and $Y_{t'<t}$ denote the sets of variables, $X_{t'}$ and $Y_{t'}$, taken at earlier time points $t'$ than $t$.

While **Equation 1** is asymmetric upon $X/Y$ permutation, a simple comparison of Transfer Entropy asymmetry (e.g., $T_{X \to Y} > T_{Y \to X} \geqslant 0$) does not necessarily translate into causal direction as this asymmetry is also expected for non-causal relations. Interestingly, this is in fact the absence of Transfer Entropy in one direction (e.g., $T_{Z \to X} \approx 0$), which suggests the possibility of a causal relation in the opposite direction, $X \to Z$, as in the case of v-structures in graph-based causal discovery methods, provided that a latent common cause can be excluded between the two variables (as discussed above).

We clarify in Theorem 1 this relation between temporal causality without reference to any structural model (**Equation 1**) and structural causality entailed by time-unfolded causal graphical models ($\mathcal{G}_t$). This highlights the common foundations of temporal and structural causalities beyond their seemingly unrelated definitions.

**Theorem 1**. [$T_{Y \to X} = 0$ implies temporal (2-variable + time) v-structures]

If $X_t$ is adjacent to $Y_t$ in $\mathcal{G}_t$ and $T_{Y \to X} = I(X_t; Y_{t'<t} | X_{t'<t}) = 0$, then for all $Y_{t'}$ adjacent to $Y_t$ in $\mathcal{G}_t$, with $t' < t$, there is a temporal (2-variable + time) v-structure, $Y_{t'} \to Y_t \leftarrow X_t$, in $\mathcal{G}_t$ (**Figure 2a**).

**Proof**: If $T_{Y \to X} = I(X_t; Y_{t'<t} | X_{t'<t}) = 0$, then all pairs $(X_t, Y_{t'})$ should be unconnected (assuming 'faithfulness', i.e., no coincidental cancellation of effects) and all unshielded triples $Y_{t'} - Y_t - X_t$ should be temporal v-structures, $Y_{t'} \to Y_t \leftarrow X_t$, as $Y_t \notin X_{t'<t}$ in $T_{Y \to X} = I(X_t; Y_{t'<t} | X_{t'<t}) = 0$. □

Theorem 1 can be readily extended to include the presence of other observed variables, $V_{t'\leqslant t}$, by redefining Transfer Entropy as $T_{Y \to X} = I(X_t; Y_{t'<t} | X_{t'<t}, V_{t'\leqslant t})$, which discards contributions from indirect paths through other observed variables, $V_{t'\leqslant t}$.

Note, however, that the converse of Theorem 1 is not true: a temporal v-structure does not imply a vanishing Transfer Entropy, as shown with the counterexample in *Figure 2b*. As a result, the presence of a temporal v-structure, $Y_{t'} \rightarrow Y_t \leftarrow X_t$ in $\mathcal{G}_t$, does not necessarily imply a vanishing Transfer Entropy, $T_{Y \rightarrow X} = 0$, as long as there remains an edge between any $Y_{t''}$ and $X_t$, as in the example in *Figure 2b*. Hence, Granger–Schreiber causality is in fact too restrictive and may miss actual causal effects, which can be uncovered by structural causal discovery methods like tMIIC. In addition, Granger–Schreiber causality is also known to infer spurious causal associations by excluding the presence of latent common causes a priori. By contrast, tMIIC includes time-delayed as well as synchronous effects originating from unobserved latent variables, as discussed above.

## Benchmarking of CausalXtract's causal discovery module (tMIIC)

The performance of CausalXtract's causal discovery module (tMIIC) has been assessed using Tigramite package (*Runge, 2020*), which provides different methods to learn temporal causal networks from time-series data. We compared tMIIC to two methods capable of orienting contemporaneous edges (PC and PCMCI+) and tested three different kernels for estimating mutual information (Parcorr, GPDC, and KNN). Benchmark networks and datasets have been chosen to resemble the real-world data analyzed in this study (i.e., similar number of time steps, network size, and degree distribution) and include a large range of linear and nonlinear relations between variables.

A first series of datasets was generated for a 15-node benchmark network (*Figure 2—figure supplement 1a*) with linear combinations of contributions inspired by the Tigramite package (Appendix 1). Running times and scores (Precision, Recall, Fscore) have been averaged over 10 datasets (*Figure 2—figure supplement 1b*) and show that tMIIC scores are at par with PC and PCMCI+ using GPDC or KNN kernels but that tMIIC runs orders of magnitude faster, which enables to use tMIIC on much larger datasets of biological interest including a few tens or hundreds of thousands samples. Only PC or PCMCI+ using ParCorr kernel match tMIIC running speed but with significantly lower scores, as Fscores level off around 0.6–0.7 at large sample size, while tMIIC Fscore exceeds 0.9 (*Figure 2—figure supplement 1b*).

Importantly, increasing the number of time-lagged layers from $\tau = 2$ (as in the actual model) to 5 or 10 layers in the inferred time-unfolded network (*Figure 2—figure supplement 2*) leads to very similar network reconstructions for simulated stationary data. This demonstrates tMIIC insensitivity to an overestimated maximum lag for the reconstituted network. Interestingly, however, when the generated data is no longer stationary, increasing the number of layers leads to multiple self-loops at nonstationary variables, whilst the rest of the network remains relatively unaffected (*Figure 2—figure supplement 3*). It demonstrates that CausalXtract's causal discovery module is robust to the presence of nonstationary variables but requires long-time range interactions, and therefore multiple time-lagged layers, to account for these nonstationary dynamics at specific variables. This striking observation on benchmark networks is also consistent with the multiple self-loops observed for a number of nonstationary variables in the real-world application on cellular ecosystems (*Figure 3a*, *Figure 1—figure supplement 1*).

A second series of more complex datasets was also generated for another 15-node benchmark network (*Figure 2—figure supplement 4a*) with nonlinear combinations of contributors (Appendix 2). Here, tMIIC tends to outperform both PC and PCMCI+ in terms of Recall and Fscores, while remaining orders of magnitude faster compared to GPDC and KNN kernels. Only PC or PCMCI+ using ParCorr kernel match tMIIC running speed but with significantly lower scores (i.e., Fscores level off around 0.4–0.5 at large sample size, while tMIIC Fscore exceeds 0.8). This demonstrates that CausalXtract's causal discovery module (tMIIC) is both a reliable and scalable method to discover complex temporal causal relations in very large time-series datasets including a few hundred thousand samples.

## Code availability

The source code of CausalXtract is available at https://github.com/miicTeam/CausalXtract, copy archived at *miicTeam, 2024*. It includes a demo R markdown notebook of CausalXtract pipeline, which reproduces step-by-step the results reported in the article (*Figure 3*), starting from the original live-cell time-lapse images of the tumor-on-chip ecosystem (*Figure 1a*). The Tigramite package used for benchmark comparison is available at https://github.com/jakobrunge/tigramite, copy archived at *Runge, 2024*.

## Acknowledgements

This work was supported by ITMO Cancer (grant no. 20CM106) and the European Union's Horizon 2020 research and innovation program under the Marie Skłodowska-Curie grant agreement no. 847718. LD acknowledges support from AMX PhD fellowship, VC from ARC foundation, and NL from CNRS-Imperial College joint PhD program.

## Additional information

### Funding

| Funder | Grant reference number | Author |
|---|---|---|
| Institut National de la Santé et de la Recherche Médicale | ITMO Cancer (20CM106) | Herve Isambert |
| HORIZON EUROPE Marie Sklodowska-Curie Actions | 847718 | Herve Isambert |
| École Polytechnique, Université Paris-Saclay | AMX PhD fellowship | Louise Dupuis |
| Fondation ARC pour la Recherche sur le Cancer | 4th year PhD fellowship | Vincent Cabeli |
| Centre National de la Recherche Scientifique | CNRS-Imperial College joint PhD programme | Herve Isambert |

The funders had no role in study design, data collection and interpretation, or the decision to submit the work for publication.

### Author contributions

Franck Simon, Conceptualization, Software, Validation, Investigation, Visualization, Methodology, Writing – original draft; Maria Colomba Comes, Tiziana Tocci, Data curation, Software, Validation, Investigation, Visualization, Methodology, Writing – original draft; Louise Dupuis, Resources, Software, Visualization; Vincent Cabeli, Resources, Software, Methodology; Nikita Lagrange, Resources; Arianna Mencattini, Resources, Software, Supervision, Validation, Investigation, Methodology; Maria Carla Parrini, Resources, Validation, Visualization; Eugenio Martinelli, Conceptualization, Resources, Software, Supervision, Investigation, Methodology, Writing – original draft; Herve Isambert, Conceptualization, Formal analysis, Supervision, Funding acquisition, Validation, Investigation, Visualization, Methodology, Writing – original draft, Writing – review and editing

### Author ORCIDs

Franck Simon (ID) https://orcid.org/0000-0002-6952-0819
Maria Carla Parrini (ID) https://orcid.org/0000-0002-7082-9792
Herve Isambert (ID) https://orcid.org/0000-0001-9638-8545

Reviewer #1 (Public review): https://doi.org/10.7554/eLife.95485.3.sa1
Reviewer #2 (Public review): https://doi.org/10.7554/eLife.95485.3.sa2
Author response https://doi.org/10.7554/eLife.95485.3.sa3

## Additional files

### Supplementary files
MDAR checklist

### Data availability
The original live-cell time-lapse image data and extracted crops are available at: https://doi.org/10.5281/zenodo.7755699.

The following dataset was generated:

| Author(s) | Year | Dataset title | Dataset URL | Database and Identifier |
|---|---|---|---|---|
| Parrini MC | 2023 | CausalXtract: a flexible pipeline to extract causal effects from live-cell time-lapse imaging data | https://doi.org/10.5281/zenodo.7755699 | Zenodo, 10.5281/zenodo.7755699 |

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

## Appendix 1

This appendix contains the mathematical details of the 15-node model with linear combinations of variables.

## Nodes

$$X_t^1 \leftarrow -0.47\,f_2(X_{t-1}^1) + 0.29\,f_3(X_{t-1}^2) \times \eta_1$$
$$X_t^2 \leftarrow 0.49\,f_2(X_{t-1}^2) + 0.4\,f_1(X_{t-2}^1) + \eta_2$$
$$X_t^3 \leftarrow 0.56\,f_1(X_{t-1}^3) + 0.44\,f_4(X_{t-2}^4) - 0.26\,f_2(X_{t-2}^{10}) + 0.56\,f_2(X_t^4) + \eta_3$$
$$X_t^4 \leftarrow 0.24\,f_3(X_{t-1}^4) - 0.24\,f_2(X_{t-2}^6) - 0.12\,f_4(X_{t-1}^{14}) \times \eta_4$$
$$X_t^5 \leftarrow -0.39\,f_3(X_{t-1}^5) - 0.42\,f_3(X_{t-2}^5) - 0.39\,f_3(X_t^{11}) + \eta_5$$
$$X_t^6 \leftarrow -0.32\,f_2(X_{t-1}^6) + \eta_6$$
$$X_t^7 \leftarrow -0.17\,f_4(X_{t-1}^7) - 0.17\,f_1(X_{t-2}^7) + \eta_7$$
$$X_t^8 \leftarrow 0.39\,f_4(X_{t-1}^8) - 0.46\,f_4(X_{t-1}^7) - 0.39\,f_3(X_{t-1}^1) - 0.4\,f_3(X_{t-2}^{12}) + \eta_8$$
$$X_t^9 \leftarrow -0.34\,f_1(X_{t-1}^9) + 0.43\,f_3(X_{t-2}^{12}) + \eta_9$$
$$X_t^{10} \leftarrow 0.2\,f_1(X_{t-1}^{10}) + 0.18\,f_4(X_{t-2}^9) + 0.17\,f_1(X_{t-1}^9) + 0.48\,f_3(X_{t-1}^7) - 0.26\,f_4(X_{t-1}^4) + \eta_{10}$$
$$X_t^{11} \leftarrow 0.41\,f_2(X_{t-1}^{11}) + 0.54\,f_3(X_t^2) - 0.55\,f_2(X_t^{12}) + \eta_{11}$$
$$X_t^{12} \leftarrow -0.45\,f_2(X_{t-1}^{12}) - 0.43\,f_4(X_{t-2}^3) - 0.17\,f_4(X_{t-2}^9) \times \eta_{12}$$
$$X_t^{13} \leftarrow 0.45\,f_3(X_{t-1}^{13}) + \eta_{13}$$
$$X_t^{14} \leftarrow 0.28\,f_2(X_{t-1}^{14}) + 0.37\,f_1(X_{t-2}^{12}) \times \eta_{14}$$
$$X_t^{15} \leftarrow 0.52\,f_3(X_{t-1}^{15}) + \eta_{15}$$

## Functions

$$f_1(x) = x$$
$$f_2(x) = x\,(1 - 4\,e^{-\frac{x^2}{2}})$$
$$f_3(x) = x\,(1 - 4\,x^3\,e^{-\frac{x^2}{2}})$$
$$f_4(x) = \cos(x)$$

## Noises

The $\eta$ are white noises generated for each node or contribution using a normal distribution: $\eta \sim \mathcal{N}(0, 1)$

## Appendix 2

This appendix contains the mathematical details of the 15-node model with nonlinear combinations of variables.

## Nodes

$$X_t^1 \leftarrow \eta - 0.7\,f_6(u(\eta + X_{t-1}^1)) - 0.87\,f_5(u(\eta + (X_{t-1}^{14} \times X_{t-2}^1)))$$

$$X_t^2 \leftarrow \eta + 0.65\,f_1(u(\eta + X_{t-1}^2)) - 0.63\,f_3(u(\eta + X_{t-2}^2)) + 0.79\,f_3(u(\eta + X_{t-1}^5))$$

$$X_t^3 \leftarrow \eta - 0.76\,f_5(u(\eta + X_{t-1}^3)) - 0.59\,f_6(u(\eta + X_{t-1}^7)) - 0.85\,f_2(u(\eta + X_{t-1}^{15}))$$
$$\qquad - 0.89\,f_5(u(\eta + (X_{t-2}^{13} \times X_{t-1}^7)))$$

$$X_t^4 \leftarrow \eta - 0.7\,f_6(u(\eta + X_{t-1}^5)) - 0.86\,f_2(u(\eta + X_{t-2}^8)) + 0.53\,f_1(u(\eta + (X_{t-1}^4 \times X_{t-2}^9)))$$

$$X_t^5 \leftarrow \eta + 0.54\,f_2(u(\eta + (X_{t-1}^{14} \times X_{t-2}^6)))$$

$$X_t^6 \leftarrow \eta - 0.85\,f_2(u(\eta + X_{t-1}^6)) - 0.79\,f_3(u(\eta + X_{t-2}^3)) + 0.59\,f_1(u(\eta + X_{t-1}^4))$$
$$\qquad + 0.75\,f_3(u(\eta + X_t^1)) + 0.57\,f_2(u(\eta + X_{t-1}^{14}))$$

$$X_t^7 \leftarrow \eta + 0.74\,f_1(u(\eta + X_{t-1}^7)) + 0.54\,f_6(u(\eta + X_{t-1}^9)) - 0.53\,f_2(u(\eta + (X_{t-1}^9 \times X_{t-1}^7)))$$

$$X_t^8 \leftarrow \eta \times (-0.63\,f_1(u(\eta + X_{t-1}^6)) + 0.81\,f_5(u(\eta + X_t^{13})) + 0.53\,f_6(u(\eta + (X_{t-2}^6 \times X_{t-1}^6)))$$
$$\qquad - 0.69\,f_6(u(\eta + (X_t^{13} \times X_{t-1}^6))))$$

$$X_t^9 \leftarrow \eta + 0.79\,f_3(u(\eta + X_{t-2}^4)) + 0.69\,f_6(u(\eta + (X_{t-1}^9 \times X_{t-1}^{15})))$$

$$X_t^{10} \leftarrow \eta + 0.54\,f_6(u(\eta + X_{t-1}^{10}))$$

$$X_t^{11} \leftarrow \eta + 0.83\,f_6(u(\eta + X_{t-1}^{11})) - 0.76\,f_4(u(\eta + X_{t-1}^{13})) - 0.73\,f_3(u(\eta + X_{t-1}^2))$$
$$\qquad + 0.74\,f_2(u(\eta + X_t^4)) - 0.87\,f_2(u(\eta + X_{t-2}^{10})) + 0.72\,f_4(u(\eta + X_{t-1}^{12}))$$
$$\qquad - 0.73\,f_1(u(\eta + (X_{t-2}^{10} \times X_{t-1}^{13})))$$

$$X_t^{12} \leftarrow \eta + 0.7\,f_3(u(\eta + X_{t-1}^{10})) - 0.55\,f_5(u(\eta + X_t^9)) - 0.54\,f_5(u(\eta + (X_{t-1}^{12} \times X_{t-1}^{10})))$$

$$X_t^{13} \leftarrow \eta - 0.62\,f_3(u(\eta + X_{t-2}^{14})) - 0.61\,f_1(u(\eta + (X_{t-1}^{13} \times X_{t-2}^{14})))$$

$$X_t^{14} \leftarrow \eta - 0.78\,f_6(u(\eta + X_{t-1}^{14}))$$

$$X_t^{15} \leftarrow \eta - 0.68\,f_4(u(\eta + X_{t-1}^{15})) + 0.85\,f_4(u(\eta + X_{t-2}^{15})) - 0.6\,f_5(u(\eta + X_{t-2}^{10}))$$
$$\qquad + 0.68\,f_6(u(\eta + X_{t-1}^{14})) + 0.81\,f_4(u(\eta + (X_{t-1}^{14} \times X_{t-2}^{10})))$$

## Functions

$$u(x) = \max(-1, \min(1, x))$$

$$f_1(x) = x$$

$$f_2(x) = x\,(1 - 4\,e^{-\frac{x^2}{2}})/1.52387$$

$$f_3(x) = 4\,x^2$$

$$f_4(x) = 8\,x^3$$

$$f_5(x) = 16\,x^4$$

$$f_6(x) = \cos(\pi x)$$

## Noises

The $\eta$ are white noises generated for each node or contribution using a normal distribution: $\eta \sim \mathcal{N}(0, 0.1)$.

