## [Editor Report · eLife Assessment]

This **important** study represents a data processing pipeline to discover causal interactions from time-lapse imaging data and **convincingly** illustrates it on a challenging application for the analysis of tumor-on-chip ecosystem data. The authors describe the raw data they used (imaging data), go through a step-by-step description of how to extract the features they are interested in from the raw data, and how to perform the causal discovery process. This article tackles the problem of learning causal interactions from temporal data, which is applicable to many biological applications.

---

## [Referee Report · Reviewer #1 (Public review)]

Summary:

This paper presents a data processing pipeline to discover causal interactions from time-lapse imaging data and convincingly illustrates it on a challenging application for the analysis of tumor-on-chip ecosystem data.

The core of the discovery module is the original tMIIC method of the authors, which is shown in supplementary material to compare favourably to two state-of-the-art methods on synthetic temporal data on a 15 nodes network.

Strengths:

This paper tackles the problem of learning causal interactions from temporal data which is an open problem in presence of latent variables.

The core of the method tMIIC of the authors is nicely presented in connection to Granger-Schreiber causality and to the novel graphical conditions used to infer latent variables and based on a theorem about transfer entropy.

tMIIC compares favourably to PC and PCMCI+ methods using different kernels on synthetic datasets generated from a network of 15 nodes.

A full application to tumor-on-chip cellular ecosystems data including cancer cells, immune cells, cancer-associated fibroblasts, endothelial cells and anti cancer drugs, with convincing inference results with respect to both known and novel effects between those components and their contact.

The code and dataset are available online for the reproducibility of the results.

Weaknesses:

The references to "state-of-the-art methods" concerning the inference of causal networks should be more precise by giving citations in the main text, and better discussed in general terms, both in the first section and in the section of presentation of CausalXtract. It is only in the legend of the figures of the supplementary material that we get information.

Of course, comparison on our own synthetic datasets can always be criticized but this is rather due to the absence of a common benchmark in this domain compared to other domains. I recommend the authors to explicitly propose their datasets made accessible in supplementary material as benchmark for the community.

Comments on revisions:

This is a very nice paper.

---

## [Referee Report · Reviewer #2 (Public review)]

Summary:

The authors propose a methodology to perform causal (temporal) discovery. The approach appears to be robust and is tested in the different scenarios: one related to live-cell imaging data, and another one using synthetic (mathematically defined) time series data. They compare the performance of their findings against another well-known method by using metrics like F-score, precision and recall,

Strengths:

--Performance, robustness, the text is clear and concise, The authors provide the code to review.

Comments on revisions:

The authors have addressed my concerns properly providing the needed explanations.

---

## [Author Response]

The following is the authors’ response to the original reviews.

**Reviewer #1 (Public review):**
Summary:This paper presents a data processing pipeline to discover causal interactions from time-lapse imaging data, and convicingly illustrates it on a challenging application for the analysis of tumor-on-chip ecosystem data. The core of the discovery module is the original tMIIC method of the authors, which is shown in supplementary material to compare favourably to two state-of-the-art methods on synthetic temporal data on a 15 nodes network.Strengths:This paper tackles the problem of learning causal interactions from temporal data which is an open problem in presence of latent variables. The core of the method tMIIC of the authors is nicely presented in connection to Granger- Schreiber causality and to the novel graphical conditions used to infer latent variables and based on a theorem about transfer entropy. tMIIC compares favourably to PC and PCMCI+ methods using different kernels on synthetic datasets generated from a network of 15 nodes. A full application to tumor-onchip cellular ecosystems data including cancer cells, immune cells, cancer-associated fibroblasts, endothelial cells and anti cancer drugs, with convincing inference results with respect to both known and novel effects between those components and their contact.The code and dataset are available online for the reproducibility of the results.

We thank Reviewer #1 for highlighting the main results and strengths of our paper, as well as, for his/her recommendations below to further improve the manuscript.

Weaknesses:The references to ”state-of-the-art methods” concerning the inference of causal networks should be more precise by giving citations in the main text, and better discussed in general terms, both in the first section and in the section of presentation of CausalXtract. It is only in the legend of the figures of the supplementary material that we get information. Of course, comparison on our own synthetic datasets can always be criticized but this is rather due to the absence of common benchmark and I would recommend the authors to explicitly propose their datasets as benchmark to the community.

Following Reviewer #1’s suggestion, we now compare tMIIC’s performance to other state-of-the-art causal discovery methods for time series data in the main text and in a new Figure 2. This Figure 2 also highlights the relation between graph-based causal discovery methods for time series data and Granger-Schreiber temporal causality, as discussed in more details in Methods (Theorem 1).

We also agree about the importance of sharing benchmark datasets with the community. This is the reason why we provide the dynamical equations of the 15-node benchmarks in Supplementary Tables 1 & 2, so that anyone can generate equivalent time series datasets of any desired length.

**Reviewer #2 (Public review):**
Summary:The authors propose a methodology to perform causal (temporal) discovery. The approach appears to be robust and is tested in the different scenarios: one related with live-cell imaging data, and another one using synthetic (mathematically defined) time series data. They compare the performance of their findings against another well-know method by using metrics like F-score, precision and recall,Strengths:Performance, robustness, the text is clear and concise, The authors provide the code to review.

We thank Reviewer #2 for his/her positive assessment of our work and the suggestions below to improve the manuscript.

Weaknesses:One concern could be the applicability of the method in other areas like climate, economy. For those areas, public data are available and might be interesting to test how the method performs with this kind of data.

While our main expertise concerns the analysis of biological and biomedical data, we agree that tMIIC (which is included in MIIC R package) could in principle be applied to other areas, like climate, economy.

We have not included benchmarks on such diverse types of datasets in the present manuscript, which focuses on CausalXtract’s pipeline for the analysis and causal interpretation of live-cell time-lapse imaging data from complex cellular systems.